# Hepatic Vein-Guided Approach in Laparoscopic Anatomic Liver Resection of the Ventral and Dorsal Parts of Segment 8

**DOI:** 10.3390/jpm13061007

**Published:** 2023-06-17

**Authors:** Kazuteru Monden, Hiroshi Sadamori, Toshimitsu Iwasaki, Masayoshi Hioki, Norihisa Takakura

**Affiliations:** Department of Surgery, Fukuyama City Hospital, Hiroshima 721-8511, Japan

**Keywords:** laparoscopic liver resection, hepatic vein-guided approach, anatomic liver resection, segment 8 ventral, segment 8 dorsal

## Abstract

Laparoscopic ventral and dorsal segmentectomies 8 are an option for parenchymal-sparing liver resection. However, laparoscopic anatomic posterosuperior liver segment resection is technically demanding because of its deep location and the many variations in the segment 8 Glissonean pedicle (G8). In this study, we describe a hepatic vein-guided approach (HVGA) to overcome these limitations. For ventral segmentectomy 8, liver parenchymal transection was initiated at the ventral side of the middle hepatic vein (MHV) and continued exposing it toward the periphery. The G8 ventral branch (G8vent) was identified on the right side of the MHV. Following G8vent dissection, liver parenchymal transection was completed by connecting the demarcation line and G8vent stump. For dorsal segmentectomy 8, the anterior fissure vein (AFV) was exposed peripherally. The G8 dorsal branch (G8dor) was identified on the right side of the AFV. Following G8dor dissection, the right hepatic vein (RHV) was exposed from the root. Liver parenchymal transection was completed by connecting the demarcation line and RHV. Between April 2016 and December 2022, we performed laparoscopic ventral and dorsal segmentectomy 8 in fourteen patients. No complications (Clavien–Dindo classification, Grade ≥ IIIa) were observed. An HVGA is feasible and useful for standardizing safe laparoscopic ventral and dorsal segmentectomies 8.

## 1. Introduction

In recent years, several reports have recommended parenchymal-sparing liver resection to preserve as much residual liver volume as possible, prevent postoperative liver failure, and improve repeat liver resection rates [1]. As segment 8 (S8) occupies a large volume in the liver, resection must be performed safely to preserve liver function and ensure anatomical resection, such as the resection of the ventral and dorsal parts of S8 (S8vent and S8dor, respectively), instead of segmentectomy 8 [2]. However, laparoscopic anatomic resection of the posterosuperior liver segments is technically demanding because they are surrounded by the ribs and diaphragm, forceps movement is limited, and the major hepatic veins must be exposed. Additionally, the ventral and dorsal branches of the Glissonean pedicle of S8 (G8vent and G8dor, respectively) have many variations and branches deep in the liver, making it difficult to encircle them from the hepatic hilum. We used a hepatic vein-guided approach (HVGA) to overcome these limitations. In this study, we describe the laparoscopic anatomic resection of S8vent and S8dor using the HVGA and the anatomy useful in this procedure.

### 1.1. HVGA

The hepatic veins course creates boundaries between the intersegmental or sectional planes [3,4]. The HVGA is a procedure for liver parenchymal dissection in which the hepatic veins are exposed from the root side to identify the intrahepatic Glissonean pedicle [5]. An appropriate dissection plane during liver resection can be maintained by exposing the hepatic veins [6]. Additionally, the HVGA exposes the hepatic veins from the root side, thus avoiding split injury [7].

### 1.2. Laennec’s Capsule around the Hepatic Vein

Laennec’s capsule, first described by Laennec in 1802 [8], surrounds the root of the hepatic veins and Glissonean pedicles [9]. A recent study reported that Laennec’s capsule covers both the root and peripheral side of the hepatic veins [10,11].

Additionally, a recent study identified a membrane called the cardiac Laennec’s capsule in addition to the genuine Laennec’s capsule (hereinafter referred to as the hepatic Laennec’s capsule to distinguish it from the cardiac Laennec’s capsule) [12,13]. Monden et al. reported that the cardiac Laennec’s capsule is derived from the pericardium (parietal and visceral) and diaphragm in the histopathological examinations of cadavers [14]; it covers the tunica media of the inferior vena cava (IVC) and hepatic veins. Therefore, the IVC and hepatic veins are covered by two membranes: the hepatic and cardiac Laennec’s capsules (Figure 1).

Kiguchi et al. demonstrated novel techniques using these two membranes: the inter-Laennec (Figure 2a) and outer-Laennec approaches (Figure 2b) [12,13]. These techniques can be performed by exposing the hepatic vein from the root side.

#### 1.2.1. The Inter-Laennec Approach

The inter-Laennec approach is a method of entering between the hepatic and cardiac Laennec’s capsules to expose the hepatic vein. As a result, the cardiac Laennec’s capsule is preserved on the venous side. This technique can be used when the tumor is adjacent to the hepatic vein by placing the hepatic Laennec’s capsule on the resection side to ensure a surgical margin [11,12,13]. Monden et al. demonstrated the feasibility of this technique using cadaver specimens [14].

#### 1.2.2. The Outer-Laennec Approach

The outer-Laennec approach exposes the hepatic vein while preserving both the hepatic and cardiac Laennec’s capsules on the hepatic vein side, thus maintaining the strength of the vein wall and potentially reducing blood loss [15].

#### 1.2.3. Relationship between the Right Hepatic Vein and Paracaval Branch

Kumon et al. defined the caudate lobe as consisting of the Spiegel lobe, paracavalportion, and caudate processes [16,17]. They defined the caudate lobe based on portal segmentation, and the caudate portal branches were defined as the portal vein branching from the main portal trunk or the first-order branches of the portal vein. Kumon et al. investigated the ramifications of the pattern of paracaval branches as 14/19 (73.7%) liver casts from the left portal vein and 5/19 (26.3%) from the right portal vein [17]. Regarding the extent of the area supplied by the paracaval branch, 10 of 19 (52.6%) extended beneath the diaphragm, and 6 of 19 (31.6%) extended to the anterior aspect of the IVC and dorsal aspect of the root of the major hepatic veins [17]. Maki et al. reported that in 10 of 63 (16%) healthy donor candidates, the area supplied by the paracaval branch extended to the diaphragmatic surface and ventral sides of the right hepatic vein (RHV), and 9 of 63 (14%) extended to the diaphragmatic surface and dorsal side of the RHV [18]. These results indicate that while exposing the root of the RHV, paracaval branches are frequently encountered and must be dissected (Figure 3). After exposing the middle hepatic vein (MHV), the peripheral paracaval branches can be observed by dissecting the ventral side of the IVC. Once the paracaval branch is encountered, the root of the RHV can be easily exposed by dissecting the paracaval branch. Consequently, the paracaval branch acts as a landmark to identify the root of the RHV.

#### 1.2.4. Anterior Fissure Vein

Cho et al. defined the anterior fissure vein (AFV) as the vein that courses between the ventral and dorsal part of the right anterior section (RAS) [19]. They emphasized the AFV as a landmark for the boundary between the ventral and dorsal part of the RAS [20,21].

Regarding the confluence pattern of the AFV and major hepatic veins, 84.0–90.9% flowed to the MHV, 9.1–15.1% flowed into the RHV, and 7.4% directly flowed into the IVC [20,22,23] (Figure 4). Additionally, Cho et al. reported that 28 of 44 (63.6%) patients experienced flow through the terminal portion of the MHV, 12 (27.3%) through the proximal portion of the MHV, and the remaining 4 (9.1%) into the terminal portion of the RHV [20]. As a result, the AFV enters the MHV in a high proportion of patients. If the AFV is exposed from the root of the MHV using the HVGA, the second or more branches may be the AFV; therefore, preoperative simulation is crucial.

## 2. Operative Procedures

The patient was placed in a left semi-lateral position using five trocars: a 12 mm trocar at the umbilicus for the laparoscope, three 12 mm trocars in the right subcostal arch, and a 5 mm intercostal trocar through the 8th intercostal space (Figure 5). A 5 mm tape for the Pringle maneuver was placed on the left side of the upper abdomen. The pneumoperitoneum pressure was set to 10 mmHg. The operator stood on the patient’s left side.

After dissecting the falciform and right coronary ligaments, the roots of the MHV and RHV were exposed.

### 2.1. Liver Resection of the Ventral Part of Segment 8

Following the identification and marking of the MHV using intraoperative ultrasonography (IOUS), liver parenchymal transection was initiated at the ventral aspect of the root of the MHV using the outer-Laennec approach (Figure 6a). The MHV was continuously exposed from the root to the periphery using a Cavitron ultrasonic surgical aspirator (CUSA). This exposure direction of the hepatic vein can prevent split injuries. The G8vent was identified on the right side of the MHV. Following the encircling of the G8vent (Figure 6b), it was temporally clamped, and the ischemic area was confirmed as a transection line (Figure 6c). Following G8vent dissection, the resected side of the liver was retracted upward to the right by an assistant, and liver parenchymal transection was completed by connecting the demarcation line and G8vent stump (Figure 6d). Specimens were removed through the umbilical incision and placed in a plastic bag. The MHV and G8vent stump were exposed on the cut surface. A drainage tube was placed on the cut surface.

### 2.2. Liver Resection of the Dorsal Part of Segment 8

Following the identification and marking of the AFV using IOUS, liver parenchymal transection was initiated at the ventral aspect of the root of the MHV. The confluence of the AFV and MHV was confirmed, and the AFV was continuously exposed by the CUSA from the root to the periphery (Figure 7a). The G8dor was identified at a deeper location on the right side of the AFV (Figure 7b). The G8dor was temporally clamped, and the ischemic area was confirmed to be a transection line (Figure 7c). Liver parenchymal transection was continued to the ventral side of the IVC, and the paracaval branch was dissected. Behind the dissected paracaval branch, the RHV was exposed from the root via the outer-Laennec approach (Figure 7d). Following G8dor dissection, liver parenchymal transection was completed by connecting the demarcation line and RHV. Specimens were removed through the umbilical incision and placed in a plastic bag. The AFV, RHV, and G8dor stump were exposed on the cut surface (Figure 7e). A drainage tube was placed on the cut surface.

## 3. Results

Between April 2016 and December 2022, we performed seven each of laparoscopic liver resection of the ventral and dorsal part of S8 (Table 1). The median duration of hospitalization was 6.5 days, and no patient required open surgery. According to the Clavien–Dindo classification, no complications (≥Grade IIIa) were observed. One patient had a hospital stay of 17 days and experienced a delayed recovery due to a case of cholecystitis; however, he was successfully managed with medication only (Grade Ⅱ). The presence of the AFV was confirmed by preoperative three-dimensional (3D) simulation in 13 of 14 patients (92.9%); the exception was a patient for S8vent resection. R0 resection was achieved in all the patients.

## 4. Discussion

In this study, we described the liver anatomy and techniques for safe laparoscopic liver resection of S8vent and S8dor.

The definition of the caudate lobe by Kumon et al. was supported at the Consensus Conference: Precision Anatomy for Minimally Invasive HBP Surgery (PAM-HBP Surgery) held in Tokyo in 2021, where it was defined as segment 1 [3]. Based on the pattern of the paracaval branches reported by Kumon [17], we suggest that the paracaval branch can be used as a landmark to identify the root of the RHV. If the root of the RHV cannot be exposed, it is difficult to identify the RHV, which is an intrahepatic landmark, on the cut surface. Confirming the course and extent of the paracaval branch on preoperative computed tomography (CT) and 3D simulation is important to determine its positional relationship to the RHV in advance. This knowledge is useful not only for S8dor resection but also for right anterior sectionectomy and central bisectionectomy that require exposure of the RHV.

The Laennec’s capsule around the hepatic veins consists of two layers: one is derived from the pericardium and diaphragm (cardiac Laennec’s capsule), and the other is the genuine Laennec’s capsule (hepatic Laennec’s capsule), which merge and become macroscopically indistinguishable. However, by understanding that membranes of different origins cover the hepatic veins, the two layers (outer-Laennec approach) or one layer (inter-Laennec approach) can be preserved, and the hepatic vein can be safely exposed while preserving its strength. Thus, the Laennec’s capsule concept enables the safe performance of HVGA and contributes to the standardization of the surgical procedure in anatomic liver resections.

A systematic review of landmarks to identify the segmental borders of the liver has been conducted [4]. Cho et al. [20] reported that the AFV is an anatomical landmark for the boundary between the ventral and dorsal part of the RAS. Taniai et al. described that the AFV could be useful in identifying the border between the ventral and dorsal parts of S8, although they are not easily identified [22]. In contrast, Kobayashi et al. [23] concluded that the AFV was identified in 91% of study patients, but of these only 63% functioned as a landmark; hence, they concluded that the AFV serves as a landmark in only a few patients. In our study, the AFV was confirmed in 92.9% of patients by preoperative 3D simulation and functioned as a landmark for the boundary between the ventral and dorsal parts of S8.

During anatomic liver resection, exposing the hepatic vein on the dissection plane is crucial. However, exposing and controlling the bleeding from the hepatic vein is a technically demanding procedure. In open liver resection, to prevent intraoperative massive bleeding, encircling the common trunk of the MHV and LHV, or that of the LHV for occlusion, are reported [24,25]. The Consensus Conference of PAM-HBP Surgery reported survey outcomes regarding laparoscopic liver resection [26]. The results showed that 53% of the experts “never” perform taping of the common trunk or LHV, and only 12% “often” perform taping. Encircling the common trunk itself is a risky and time-consuming procedure for laparoscopic surgery. In laparoscopic surgery, in addition to the Pringle maneuver for total hepatic inflow intermittent occlusion, hepatic vein pressure control and the direction of hepatic vein exposure are considered important. The method of exposing the hepatic vein from the root side, as in the HVGA, avoids split injury. Split injury occurs when the CUSA is moved from the peripheral side towards the root side while exposing the hepatic vein and is sometimes difficult to control. In a recent study, the relationship between airway pressure (AWP), pneumoperitoneum pressure (PPP), and central venous pressure (CVP) was examined [27]. The study findings revealed positive associations between AWP and CVP, as well as between PPP and CVP. Accordingly, lower AWP corresponded with lower CVP levels, and these authors recommended that if massive bleeding from the hepatic vein or vena cava occurs, it is often beneficial to reduce AWP or disconnect the patient from the ventilator.

Laparoscopic anatomic resection of the posterosuperior liver segments is a technically challenging procedure [28]. However, with the increasing adoption of minimally invasive techniques and the accumulation of experience, laparoscopic liver resection is now frequently performed on posterosuperior liver segments [2,29,30]. A recent systematic review revealed that for hepatocellular carcinoma (HCC) located in the posterosuperior segments, conversion to open occurred in 15.5% of patients and complications were observed in 18.6% of patients. Among these complications, 33.8% were categorized as a major [31]. These authors emphasized the necessity for experienced surgeons and high-volume centers to achieve good short-term outcomes for the laparoscopic anatomic resection of posterosuperior liver segments. In the present study, 78% of patients had HCC. Despite the high proportion of HCC cases, the HVGA was safely performed in all cases. However, this is a small sample of patients in a single institution, and we need to examine more cases to corroborate these outcomes.

## 5. Conclusions

Understanding the liver anatomy, simulating the anatomy preoperatively, and establishing exposure methods can lead to safer liver resections. The HVGA is a technically feasible and effective procedure for isolating the G8vent and G8dor located deep in the liver.

## Figures and Tables

**Figure 1 jpm-13-01007-f001:**
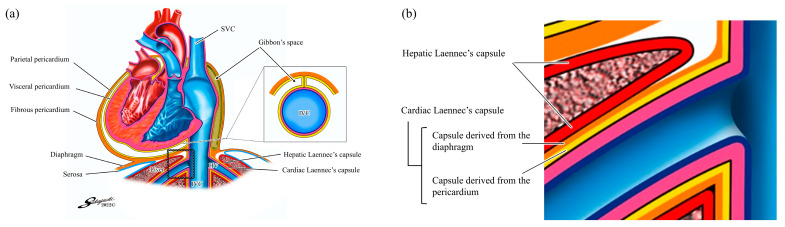
Liver anatomy based on Laennec’s capsule (adapted from reference [14]): (**a**) The cardiac Laennec’s capsule is derived from the pericardium (parietal and visceral) and diaphragm and covers the tunica media of the inferior vena cava and hepatic veins; (**b**) Enlarged image of the area within the white dotted line in (**a**). HV, hepatic vein; IVC, inferior vena cava; SVC, superior vena cava.

**Figure 2 jpm-13-01007-f002:**
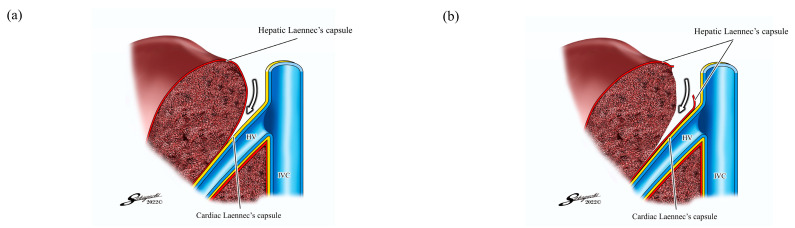
The inter-Laennec and outer-Laennec approaches to expose the major hepatic vein (adapted from references [11,12]): (**a**) The inter-Laennec approach is a method of entering between the hepatic and cardiac Laennec’s capsules to expose the hepatic vein; (**b**) The outer-Laennec approach exposes the hepatic vein while preserving both the hepatic and cardiac Laennec’s capsules on the hepatic vein side. HV, hepatic vein; IVC, inferior vena cava.

**Figure 3 jpm-13-01007-f003:**
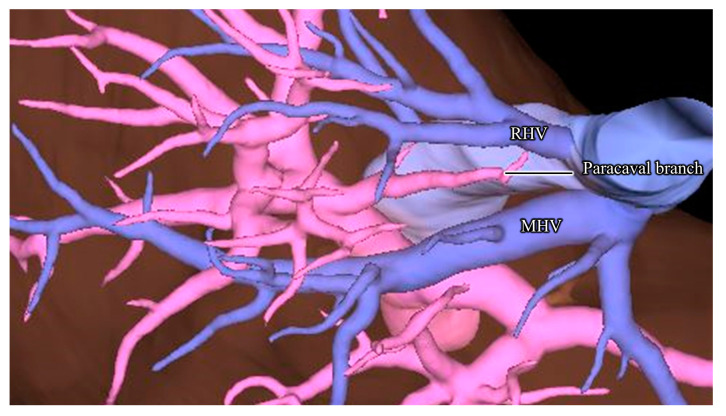
Relationship between the right hepatic vein (RHV) and paracaval branch: the paracaval branch extends beneath the diaphragm and ventral aspect of the root of RHV. LHV, left hepatic vein; MHV, middle hepatic vein.

**Figure 4 jpm-13-01007-f004:**
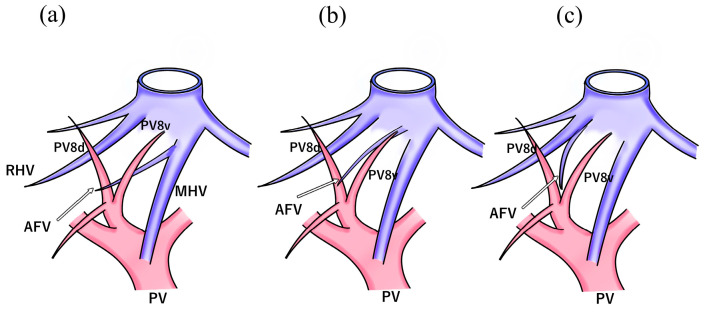
Confluence pattern of the anterior fissure vein: (**a**) 84–90.9% flowed to the middle hepatic vein; (**b**) 7.4% flowed directly into the inferior vena cava; (**c**) 9.1–15.1% flowed to the right hepatic vein. AFV, anterior fissure vein; IVC, inferior vena cava; LHV, left hepatic vein; MHV, middle hepatic vein; PV, portal vein; PV8d, dorsal branch of the portal vein of segment 8; PV8v, ventral branch of the portal vein of segment 8; RHV, right hepatic vein.

**Figure 5 jpm-13-01007-f005:**
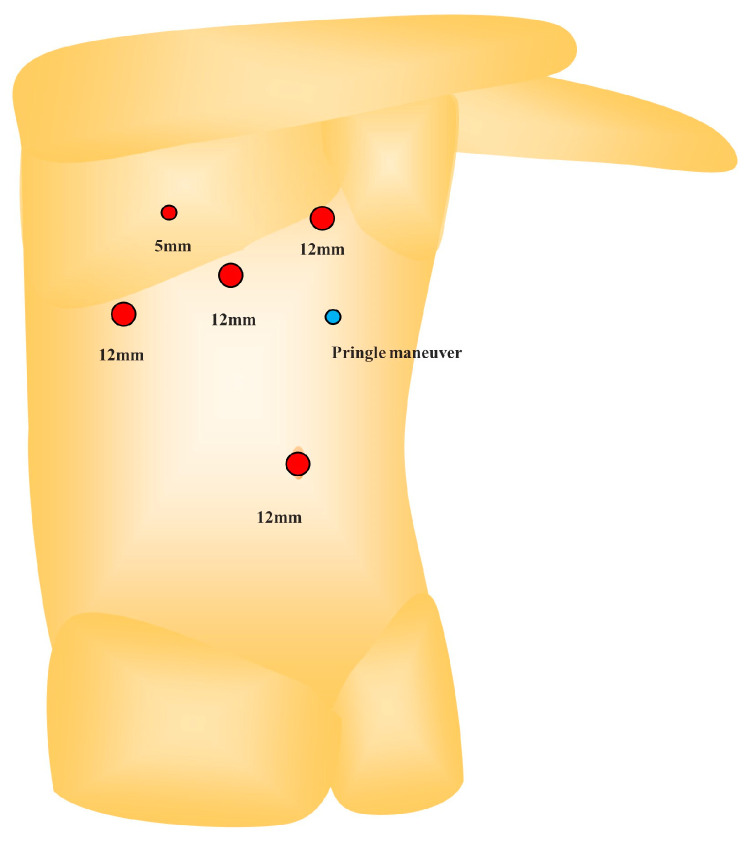
Trocar placement: The patient is placed in a left semi-lateral position. A trocar is inserted through the umbilicus for the laparoscope. Three trocars are placed below the right subcostal arch. A 5 mm intercostal trocar is placed in the 8th intercostal space.

**Figure 6 jpm-13-01007-f006:**
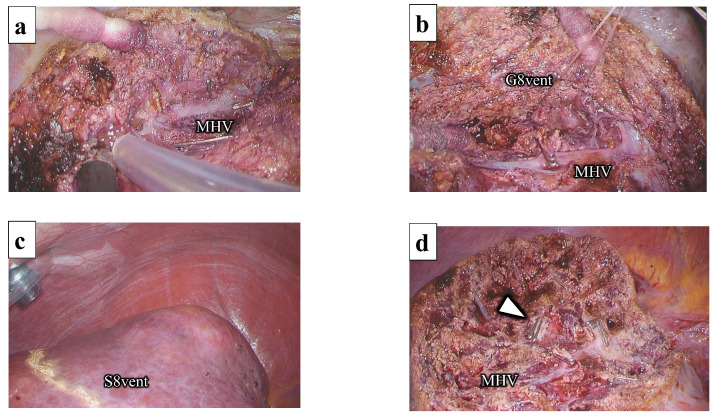
Liver resection of the ventral part of segment 8: (**a**) The MHV is exposed from the root; (**b**) The G8vent is encircled at the right side of the MHV; (**c**) The demarcation line was marked as a transection line; (**d**) The G8vent stump (arrowhead) and MHV are exposed on the cut surface. G8vent, the ventral branch of the Glissonean pedicle of segment 8; S8, segment 8; MHV, middle hepatic vein.

**Figure 7 jpm-13-01007-f007:**
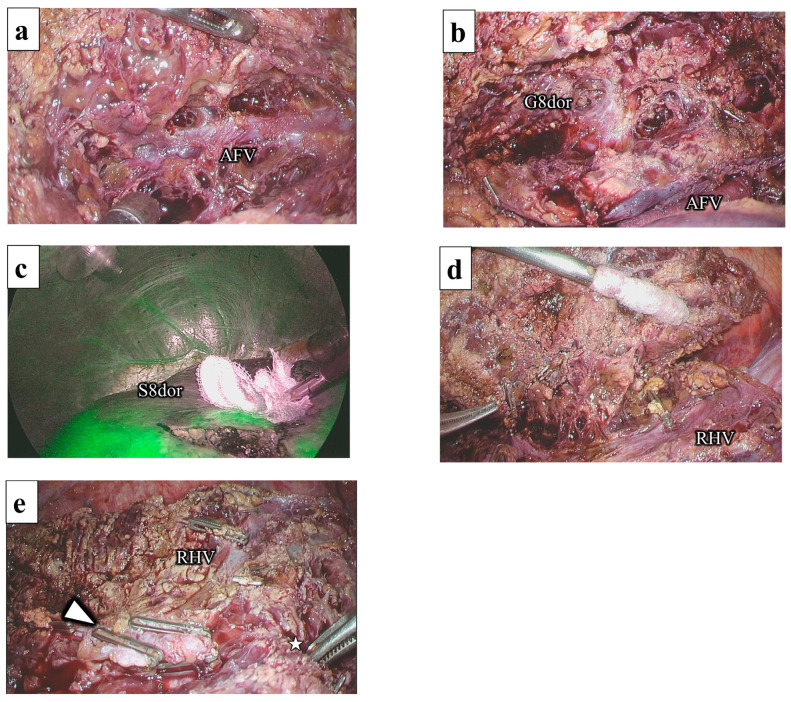
Liver resection of the dorsal part of segment 8: (**a**) The AFV is exposed from the root; (**b**) The G8dor is encircled at the right side of the AFV; (**c**) S8dor is apparent on indocyanine green fluorescence imaging; (**d**) The RHV is exposed from the root; (**e**) The G8dor stump (arrowhead), RHV, and AFV (star) are exposed on the cut surface. AFV, anterior fissure vein; G8dor, the dorsal branch of the Glissonean pedicle of segment 8; RHV, right hepatic vein; S8dor, the dorsal part of segment 8.

**Table 1 jpm-13-01007-t001:** Patient characteristics and perioperative outcomes.

Case	Surgical Procedure	AFV Confirmation in Preoperative 3D Simulation	Hepatitis Status	Child–Pugh Classification	Diagnosis	Operative Time, Min	Blood Loss, mL	Resection Status (%)	Complication	Length of Hospital Stay (Days)
1	S8vent resection	**+**	HBV	A	HCC	448	0	R0	None	10
2	S8vent resection	**+**	−	A	HCC	290	135	R0	None	5
3	S8vent resection	**+**	−	A	CRLM	225	0	R0	None	5
4	S8vent resection	**+**	−	A	HCC	337	80	R0	None	5
5	S8vent resection	−	HBV	A	HCC	298	50	R0	None	9
6	S8vent resection + left medial sectionectomy	**+**	HBV	A	HCC	349	20	R0	None	4
7	S8vent + partial resections	**+**	−	A	CRLM	340	100	R0	None	7
8	S8dor resection	**+**	HBV	A	HCC	319	0	R0	None	6
9	S8dor resection	**+**	HCV	A	HCC	347	50	R0	None	7
10	S8dor resection	**+**	−	A	HCC	474	290	R0	None	17
11	S8dor resection	**+**	HCV	A	HCC	322	400	R0	None	8
12	S8dor resection	**+**	HCV	A	HCC	385	60	R0	None	6
13	S8dor resection	**+**	−	A	HCC	384	0	R0	None	6
14	S8dor resection + partial resections	**+**	−	A	CRLM	337	200	R0	None	7
**Median**						338.5 (225–474)	55 (0–400)			6.5 (4–17)

Data are reported as median (range). 3D, three-dimensional; AFV, anterior fissure vein; CRLM, colorectal liver metastasis; HBV, hepatitis B virus; HBC, hepatitis C virus; HCC, hepatocellular carcinoma; S8dor, dorsal part of segment 8; S8vent, ventral part of segment 8.

## Data Availability

Data is unavailable due to privacy or ethical restriction.

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
