# Peer review of "Hepatic Vein-Guided Approach in Laparoscopic Anatomic Liver Resection of the Ventral and Dorsal Parts of Segment 8"

_jpm, 2023, doi:10.3390/jpm13061007_

Round 1

Reviewer 1 Report

Very interesting study, describing difficult anatomical approach through minimally invasive methods. This paper would merit hepatic surgeons specialising in laparoscopic liver resection

Author Response

RESPONSES TO COMMENTS FROM THE REVIEWERS

Reviewer 1
Comments
Very interesting study, describing difficult anatomical approach through minimally invasive methods. This paper would merit hepatic surgeons specialising in laparoscopic liver resection

Response: Thank you for your kind feedback on our study describing the challenging anatomical approach through minimally invasive methods. We greatly appreciate your recognition of the potential relevance of our findings for liver surgeons specializing in laparoscopic liver resection.

Reviewer 2 Report

The authors describe a retrospective study focused on laparoscopic liver resection of ventral and dorsal parts of segment VIII proposing a new technical approach defined "hepatic vein-guided approach" (HVGA). The topic is of great interest and is original. The anatomical description in the background provide a punctual definition of the main landmarks for understanding the approach. The statistical design is suitable for this type of technical description and sample size is 14 patients. The English is fine. The operative procedure Is well described. The results and discussion could be improved. Conclusion are adeguate.

There are some issue to be solved:

1) There is a mistake in the manuscript about the post operative morbidity. In fact the authors described in the abstract (line 19 page 1) that "...no complications CD ≥IIIa were observed..." and in the results (line 201 page 6) "...no complications CD ≤IIIa were observed..". The authors need to solve this inconsistency. Also they have to clarify about post operative morbidity, in fact also in the table 1; they declare that no complications occurred, but there are great difference between hospital stay ( patient n 10 rest 17 days ). So the authors need to better specify about morbidity dividing CD grade </> IIIa. 

2) The discussion could be enriched with some interesting discussion points that should be pointed out: 

- this approach is very interesting and very safe not only for SVIII ventral and dorsal resection, but also for other high risk or high difficult procedure (i.e. central hepatectomy). Infact, the vascular control is crucial in this type of procedure as the intraoperative hemorrhage could be fatal. About that I advice the authors to include in the manuscript this two paper that described technical strategy to prevent intraoperative hepatic massive bleeding and insert a paragraph to discuss the importance of vascular control in liver resection : 1) Giuliante F, Nuzzo G, Ardito F, Vellone M, De Cosmo G, Giovannini I. Extraparenchymal control of hepatic veins during mesohepatectomy. J Am Coll Surg. 2008;206(3):496-502. 2) Muttillo EM, Felli E, Cinelli L, Giannone F, Felli E. The counterclock-clockwise approach for central hepatectomy: A useful strategy for a safe vascular control. J Surg Oncol. 2022 Feb;125(2):175-178. doi: 10.1002/jso.26707.  PMID: 34609000.

2) The authors describe a very interesting procedure but also it's important to discuss some information about the learning curve. In fact as described poster-superior segments tumors are very difficult level of hepatic resection. Could be interesting discuss how the laparoscopic approach could help in finding landmarks better in postero superior segments. I suggest to adding this recent systematic review on difficult scenarios In laparoscopic liver surgery that show some important finding about learning curve and results in laparoscopic liver resection for hcc located in postern-superior segments.Berardi G, Muttillo EM, Colasanti M, Mariano G, Meniconi RL, Ferretti S, Guglielmo N, Angrisani M, Lucarini A, Garofalo E, Chiappori D, Di Cesare L, Vallati D, Mercantini P, Ettorre GM. Challenging Scenarios and Debated Indications for Laparoscopic Liver Resections for Hepatocellular Carcinoma. Cancers (Basel). 2023 Feb 27;15(5):1493. doi: 10.3390/cancers15051493. PMID: 36900284; PMCID: PMC10001345. 

Author Response

Reviewer 1
Comments
Very interesting study, describing difficult anatomical approach through minimally invasive methods. This paper would merit hepatic surgeons specialising in laparoscopic liver resection

Response: Thank you for your kind feedback on our study describing the challenging anatomical approach through minimally invasive methods. We greatly appreciate your recognition of the potential relevance of our findings for liver surgeons specializing in laparoscopic liver resection.

Reviewer 2

Comments

1.  There is a mistake in the manuscript about the post operative morbidity. In fact the authors described in the abstract (line 19 page 1) that "...no complications CD ≥IIIa were observed..." and in the results (line 201 page 6) "...no complications CD ≤IIIa were observed.."

 Also they have to clarify about post operative morbidity, in fact also in the table 1; they declare that no complications occurred, but there are great difference between hospital stay ( patient n 10 rest 17 days ). 

Response:

Thank you for pointing out the error in our manuscript regarding the description of post-operative morbidity. We apologize for the confusion caused. After reviewing the manuscript, we have identified the mistake in the results section.

We corrected the results section.

(line 211-214, page 6).

According to the Clavien-Dindo classification, no complications (≥ Grade IIIa) were observed. One patient had a hospital stay of 17 days and experienced a delayed recovery due to a case of cholecystitis; however, he was successfully managed with medication only (Grade Ⅱ).

  1. I advise the authors to include in the manuscript this two paper that described technical strategy to prevent intraoperative hepatic massive bleeding and insert a paragraph to discuss the importance of vascular control in liver resection

Response:

Vascular control is critical, as intraoperative hemorrhage can have fatal consequences. I appreciate your suggestion to include two relevant papers in the manuscript:

These papers describe technical strategies to prevent intraoperative hepatic massive bleeding and could contribute to the discussion on the importance of vascular control in liver resection.

We have added this information to the Discussion section:

(line258-277, page 8-9).

During anatomic liver resection, exposing the hepatic vein on the dissection plane is crucial. However, exposing and controlling the bleeding from the hepatic vein is a technically demanding procedure. In open liver resection, to prevent intraoperative massive bleeding, encircling the common trunk of the MHV and LHV, or that of the LHV for occlusion, are reported [24, 25]. The Consensus Conference of PAM-HBP Surgery reported sur-vey outcomes regarding laparoscopic liver resection [26]. The results showed that 53% of the experts “never” perform taping the common trunk or LHV, and only 12% “often” per-form taping. Encircling the common trunk itself is a risky and time-consuming procedure for laparoscopic surgery. In laparoscopic surgery, in addition to the Pringle maneuver for total hepatic inflow intermittent occlusion, hepatic vein pressure control and the direction of hepatic vein exposure are considered important. The method of exposing the hepatic vein from the root side, as in the HVGA, avoids split injury of the hepatic vein. Split injury occurs when the CUSA is moved from the peripheral side towards the root side while ex-posing the hepatic vein and is sometimes difficult to control. In a recent study, the relationship between airway pressure (AWP), pneumoperitoneum pressure (PPP), and central venous pressure (CVP) was examined [27]. The study findings revealed positive associations between AWP and CVP, as well as between PPP and CVP. Accordingly, lower AWP corresponded with lower CVP levels, and these authors recommended that if massive bleeding from the hepatic vein or vena cava occurs, it is often beneficial to reduce AWP or disconnect the patient from the ventilator.

  1. I suggest to adding this recent systematic review on difficult scenarios In laparoscopic liver surgery that show some important finding about learning curve and results in laparoscopic liver resection for HCC located in postern-superior segments.

Response:

Thank you for your comment. We have added this information to the Discussion section:

(line 278-288, page 9).

Laparoscopic anatomic resection of the posterosuperior liver segments is a technically challenging procedure [28]. However, with the increasing adoption of minimally invasive techniques and the accumulation of experience, laparoscopic liver resection is now frequently performed on posterosuperior liver segments [29–31]. A recent systematic re-view revealed that for hepatocellular carcinoma (HCC) located in the posterosuperior segments, conversion to open occurred in 15.5% of patients and complications were ob-served in 18.6% of patients. Among these complications, 33.8% were categorized as a ma-jor [32]. These authors emphasized the necessity for experienced surgeons and high-volume centers to achieve good short-term outcomes for the laparoscopic anatomic resection of posterosuperior liver segments. In the present study, 78% of patients had HCC. De-spite the high proportion of HCC cases, the HVGA was safely performed in all cases.

We appreciate the time and effort you and each of the reviewers have dedicated to providing insightful feedback on ways to strengthen our paper.